# Influence of Tert-Butylthiol and Tetrahydrofuran on the Holographic Characteristics of a Polymer Dispersed Liquid Crystal: A Research Line Toward a Specific Sensor for Natural Gas and Liquefied Petroleum Gas

**DOI:** 10.3390/polym11020254

**Published:** 2019-02-02

**Authors:** María P. Mora, Manuel G. Ramírez, Francisco Brocal, Manuel Ortuño, Augusto Beléndez, Inmaculada Pascual

**Affiliations:** 1Instituto Universitario de Física Aplicada a las Ciencias y las Tecnologías, Universidad de Alicante, Apartado 99, 03080 Alicante, Spain; pilarmorabeneyto@gmail.com (M.P.M.); ramirez@ua.es (M.G.R.); francisco.brocal@ua.es (F.B.); a.belendez@ua.es (A.B.); pascual@ua.es (I.P.); 2Departamento de Física, Ingeniería de Sistemas y Teoría de la Señal, Universidad de Alicante, Apartado 99, E03080 Alicante, Spain; 3Departamento de Óptica, Farmacología y Anatomía, Universidad de Alicante, Apartado 99, E03080 Alicante, Spain

**Keywords:** holography, holographic sensor, natural gas sensor, holographic polymer dispersed liquid crystal

## Abstract

Tert-Butylthiol (TBT) and tetrahydrothiophene (THT) are odorant substances added to natural gas and liquefied petroleum gas to help their detection by the human smell. In this research, TBT and THT are incorporated into a holographic polymer-dispersed liquid crystal and their influence in the main holographic characteristics of the photopolymer are studied in order to open the way towards the design of a holographic sensor to detect natural gas and liquefied petroleum gas.

## 1. Introduction

The design of sensors for qualitative and quantitative detection of different substances such as metabolites, gases, environmental hazards, ions, etc. is a research subject of great interest [1,2]. In the last few years, the possibilities of holography related to sensors continue to be explored [3,4,5]. In a hologram, any physical or chemical stimulus can produce changes in the diffraction grating spacing or in the refractive index modulation. This will result in a modification of the diffracted and transmitted beams that can be used to develop a detection method or a specific sensor.

Sensors for gas detection use different technologies such as thermal conductivity, electrochemical, optical, etc. [6,7,8,9,10,11,12]. Gaseous fuels such as natural gas (NG) and liquefied petroleum gas (LPG) are substances that require a detailed control due to their flammable characteristics. From their manufacture and transport to their use in the final destination, these gases need to be supervised continuously by means of sensors. Catalytic sensors are the most widely used for flammable gases. They are based on the variation of the electrical resistance of platinum with the temperature generated in the combustion of the gas, which is carried out catalytically by metal oxides [6]. These are non-specific sensors that produce signal with any hydrocarbon molecule.

Recently, holographic sensors for volatile hidrocarbons have been designed. The work of J. L. Martínez-Hurtado et al. [13] shows that it is possible to develop holographic sensors with adequate characteristics to detect hydrocarbons. For this idea, the next step will be to provide the holographic sensors with specificity. Thus, it will be possible to discriminate between several types of gases. For this, the internal structure of the holographic material has to be able to discriminate between different molecules. It has been shown that the use of liquid crystals (LC) in the holographic sensors can improve its response, sensitivity, and specificity [14].

Since both NG and LPG are practically odorless, mixtures of odorant substances are added which have a very low detection threshold for human smell. These are substances that contain Sulphur in its molecule. The most used belong to two different chemical groups: mercaptans and thiophene derivatives. The most representative of the mercaptans is tert-Butylthiol (TBT) and for thiophene derivatives, the most commonly used is tetrahydrothiophene (THT) (Figure 1).

TBT and THT have a certain reactivity due to their chemical structure with the presence of a Sulphur atom in their molecules. Therefore, they could influence the polymerization reactions or the molecular diffusion processes during the recording of a diffraction grating in a holographic photopolymer.

We study here the interaction of TBT and THT with the components of a holographic polymer-dispersed liquid crystal (HPDLC) in order to analyze if these substances influence any of the processes related to the photopolymer—initiation, polymerization, diffusion, etc. 

HPDLCs are made by holographic recording in a photo-polymerization induced phase separation process (PIPS) in which the liquid crystal molecules diffuse to dark areas in the diffraction grating where they remain as droplets [15,16,17,18,19]. The PIPS process involves molecular separation due to the formation of a three-dimensional polymer network by means of monomers with high functionality, usually between 3 and 6 [20]. The LC diffusion to the dark areas in the diffraction grating greatly increases the modulation of the refractive index.

We have chosen an HPDLC because it is a composite that contains LC; therefore, the possibilities of interaction with TBT and THT are greater. In addition, an HPDLC allows more alternatives for future work with the objective to develop a specific sensor.

In this work, it is shown that both TBT and THT modify the holographic characteristics of the photopolymer and this has allowed us to explore possibilities to continue the research. Moreover, we also detail the experimental procedures used, specially the preparation of the samples. However, the design of a specific holographic sensor for NG and LPG will require additional research.

## 2. Photopolymer Formulation

The monomer used was dipentaerythritol penta/hexa-acrylate (DPHPA) with a refractive index n = 1.490. We used the nematic liquid crystal QYPDLC-036 from Qingdao Intermodal Co., Ltd. (Qingdao, China) which is a mixture of 4-cyanobiphenyls with alkyl chains of different lengths. It has an ordinary refractive index n_0_ = 1.520, and a difference between extraordinary and ordinary index Δn = 0.250. *N*-methyl-2-pyrrolidone (NMP) was used as a solvent. Octanoic acid (OA) was used as a cosolvent and surfactant. We used ethyl eosin (EE) as a dye and *N*-phenyl glycine (NPG) as an initiator. Table 1 shows the composition for the two materials used. Polymer A does not contain NPG and therefore is not a true photopolymer since it does not react under the laser exposure used.

The solution was made by mixing the components under red light where the material is not sensitive. The solution was sonicated at 35 °C in an ultrasonic bath. 

We have incorporated TBT and THT to polymers A and B in order to evaluate the interactions of these substances with the polymers and the possible modification of their characteristics. Section 4.1 includes the experiments made with TBT and Section 4.2 includes the experiments made with THT.

### 2.1. Sample Preparation

Usually, optical quality glass is used as a support for the preparation of holographic photopolymer samples due to its high light transmission and therefore low losses by light diffusion. Here, we have used polyester sheets that were 100 µm thick. This material has a light transmission lower than glass but greater versatility on cutting and a lower cost. The polyester film needs to change the process for the sample preparation that now is adapted for the work with thinner and flexible layers. This aspect could be interesting for this application since gas permeable materials or membranes could be needed in a future research to obtain a sensor. The lower light transmission of the polyester films is not an inconvenience since it is not necessary to obtain an image of the reconstructed hologram, only the light intensity of the transmitted and diffracted beams (Figure 2).

A 37 × 25 mm^2^ polyester layer is placed on a filter paper with a glass base. 18µl of photopolymer solution are deposited as a line on the center of the polyester layer along the larger axis by means of a volumetric pipette. The solution is covered with other set of polyester-paper-glass base forming a sandwich which is subjected to a pressure of 2000 Pa during 300 s. Figure 3 includes a diagram of the sandwich formed by the different materials.

The preparation of the samples requires careful handling and previous training to ensure that the solution is compressed uniformly on the surface of the polyester layer.

Subsequently, the glass supports are removed and the filter papers are observed to verify if they have photopolymer solution. The samples in which the exterior paper is stained are discarded. The remaining samples are observed under red light to discard those in which the photopolymer is not distributed homogeneously on the surface (Figure 4). 

The thoroughness in the preparation of the samples is fundamental for the results to be consistent, so as to avoid erroneous interpretations. It is also necessary to control the ambient temperature since it influences the density and viscosity of the photopolymer and affects the amount of solution measured by the pipette as well as the polymerization reactions during the recording of the diffraction grating and the homogeneity of LC dispersion in the photopolymer solution (Figure 5).

## 3. Optical Set-Up

We obtained diffraction gratings with the optical set-up included in Figure 6. A Nd:YAG laser tuned at a wavelength of 532 nm with TE polarization was used to record diffraction gratings by means of continuous laser exposure. The laser beam was split into two secondary beams with an intensity ratio of 1:1. The diameter of these beams was increased to 1 cm by means of a pinhole and lenses, while spatial filtering was ensured. The object and reference beams were recombined at the sample at an angle of 16.0° to the normal with an appropriate set of mirrors, and the spatial frequency obtained was 1036 lines/mm. The working intensity at 532 nm was 2 mW/cm^2^. The diffracted and transmitted intensity were monitored in real time with a He–Ne laser positioned at Bragg’s angle (19.1°) with TE polarization and tuned to 632.8 nm, where the material does not polymerize. The recordings were made at 23 °C and the exposure time was adjusted to 40 s after a series of experiments to obtain a high diffraction efficiency without overmodulation following our previous works related to HPDLC materials [21,22].

## 4. Results

### 4.1. Experiments with TBT

TBT contains a –SH group that can intervene in redox reactions (Figure 1). In fact, eosin-Y, a derivative from fluorescein very similar to EE, has been used together with thiols and it can initiate polymerization reactions of thiol-ene type [23]. In this experiment, we test if TBT can initiate the polymerization using the polymer A that does not contain the initiator NPG.

1.5 mg of 13 µm glass hollow microspheres from Sigma-Aldrich (St. Louis, MO, USA) and 20 μL TBT are added to 500 μL polymer A in a plastic microtube. The components are mechanically mixed. 18 μL of solution are used to prepare the samples as it has been previously detailed in Section 2.1. After this, the samples are exposed to laser recording in the optical set-up.

During the recording of the diffraction grating we obtain the graph of Figure 7 which shows the diffraction efficiency (DE) versus the energetic exposure (E). DE is the diffracted light intensity divided by incident light intensity.

It can be seen that there is no diffracted beam (DE = 0%). This implies that there is no diffraction grating recorded in the photopolymer. Therefore, TBT cannot initiate the polymerization of DPHPA with the conditions used in the experiment. 

Oher thiols can initiate polymerization reactions and are used in holographic composites in which the thiol-ene mechanism has proven to be very efficient for polymerizing multifunctional monomers [24]. Therefore the result obtained implies that if TBT reacts with EE, the radical produced cannot react with the monomer. This radical may have low reactivity due to the electronic stabilization by the tert-butyl group (Figure 1).

The polymer B that contains the initiator NPG is then used, preparing a series of solutions with different concentration of TBT: 0, 1.96, 3.85, and 5.66 (Vol.%). During hologram recording, we obtain the result shown in Figure 8.

We can see that graph without TBT (TBT 0%) has the usual behavior for this photopolymer. The graph TBT 1.96 Vol.% has a change of slope that does not appear for TBT 0%. This results in lower values for DE. The graph TBT 3.85 Vol.% has a change of slope greater than graph 1.96 Vol.%; therefore, the values of DE are lower. The graph TBT 5.66 Vol.% results in small values of DE. The overall effect produced by the addition of TBT is a decrease of the DE values when the TBT concentration is increased, due to a change of slope produced in the graphs. 

Related to the energetic exposure, when the concentration of TBT increases, the change of slope occurs at lower exposure values. Thus, for 1.96 Vol.%, the change of slope occurs at 18 mJ/cm^2^ and for 3.85 Vol.%, it occurs at 12 mJ/cm^2^. The graph with 5.66 Vol.% has not change of slope because the slope is lower than in the previous graphs from DE > 0%.

Figure 9 includes the angular scan obtained in the reconstruction of the hologram.

It can be seen that the maximum diffracted efficiency (DEmax) obtained decreases drastically when the concentration of TBT is increased. Figure 10 includes the DEmax values for each concentration of TBT. 

The values decrease exponentially according to the equation included in the Figure. The DEmax is below 10% for a relatively low concentration of TBT (5.66 Vol.%).

The results obtained with the combination of polymer B and TBT show changes of slope once the registry is started. They also show a drastic reduction of DE with low TBT concentration and DEmax values that decrease exponentially with the increase of the TBT concentration, and that are close to 0% with a relatively low concentration of TBT. Therefore we can conclude that TBT acts as a retarder in this photopolymer [25]. It lowers the polymerization rate (changes of slope) and the conversion degree (decrease in DEmax).

We propose here a tentative mechanism to explain these results, taking account that TBT must have a direct influence in the free radical polymerization during the holographic recording.

The basic mechanism in a free radical polymerization has three different processes: Initiation, propagation, and termination. We take into account only the initiation reactions (1) and (2). During laser exposure, initiator radicals R are produced, which can react with the monomer to produce chain initiators M [25]:(1)$$I\overset{h\nu }{\to }2R\cdot$$
(2)$$R\cdot +M\overset{k_{i}}{\to }M_{1}\cdot$$where I is the initiator, *h**ν* indicates the energy absorbed from a 532 nm photon, and M represents a monomer molecule. *k_i_* is the kinetic constant for the initiation.

This scheme shows a type 1 initiation since the initiator is broken down directly into radicals. Otherwise, a type 2 Initiation needs a co-initiator that reacts with an excited dye producing R· radicals. The photopolymer used here develops a type 2 Initiation. The reaction between the excited dye and the electron donor (co-initiator) leads to the production of free radicals derived from co-initiator that initiate the polymerization reacting with the monomer M.

Photopolymers with dye derived from fluorescein undergo a redox reaction initiated by light. The dye absorbs light and is reduced whereas the co-initiator is oxidized, generating a free radical. This redox reaction initiated by light can be summarized as follows:(3)$$\mathrm{EE}+\mathrm{NPG}\overset{h\nu_{532}}{\to }{\mathrm{EE}}_{reduced}+\mathrm{NPG}\cdot$$

EE*_reduced_* is the reduced form of EE and it is a colorless molecule. The free radicals NPG· initiate the polymerization reacting with the monomer M. The scheme shown is not stoichiometrically adjusted. M. R. Gleeson et al. made a complete study for this type of initiation and they proposed a kinetic model that can be consulted in Ref. [26].

We have proven that TBT cannot initiate the polymerization without NPG (polymer A). Therefore, when we introduce TBT in a polymer with NPG (polymer B), the initiation takes place in the usual way, i.e., the excited dye reacts with NPG, producing a free radical derived from NPG molecule.

We propose that TBT can react with the free radical derived from NPG avoiding that it reacts with the monomer. Therefore, we have a new reaction in the kinetic diagram:(4)NPG·+tBu-SH→NPG+tBu-S·

This reaction generates a tBu-S· free radical due to the weakness of the S-H bond and the +I inductive effect generated by the tertbutyl group that stabilizes the free radical [27]. This is a free radical with low reactivity and it would not be able to react with the monomer. Therefore, the overall effect produced is the wasting of dye without generation of growing polymer chain radicals M· resulting in a low polymerization, low refractive index modulation, and a lower value of DE, according to the Kogelnik equation for volume holograms [28].

Moreover, we have obtained that a low concentration of TBT (1.96%) decreases the DEmax from 56% to 30% and a relatively low concentration lower than 6% decreases the DEmax to 6%. This, along with the changes of slope observed during the holographic recording point to a pseudo-catalytic process with TBT regeneration. A tentative scheme could be as follows:(5)$${\mathrm{EE}}^{3^{*}}+\mathrm{tBu}-S\cdot \to \mathrm{tBu}-\mathrm{SH}+{\mathrm{EE}}_{reduced}$$

This could occur if the TBT radical was reactive enough to react with excited EE, deactivating another dye molecule and even the TBT molecule could be regenerated in this process by a redox or photo-redox reaction [27].

In HPDLC polymers, the high functionality monomers allow the obtaining of a compact network and an effective PIPS with a relatively low polymerization [29]. This mechanism requires a dye concentration lower than photopolymers without LC [30,31]. Therefore, changes in the dye concentration produce an amplified effect on index modulation. This agrees with the result obtained. A relatively low concentration of TBT is enough to stop the polymerization because the dye is the substance involves in the process.

### 4.2. Experiments with THT

Different volumes of THT: 10, 20, and 30 μL are added to 500 μL of polymer A to obtain three solutions of photopolymer with these concentrations of THT: 1.96, 3.85, and 5.66 Vol.%. It must be taken into account that polymer A does not contain the initiator NPG (Table 1) and it cannot initiate a photopolymerization reaction under the laser exposure used. Therefore, it is expected that no hologram will be recorded and the diffraction efficiency DE must be zero during reconstruction, as in Figure 7.

The solutions are prepared as detailed in Section 4.1 and are subsequently exposed to laser recording in the optical set-up. After, the holograms are reconstructed and we obtain the angular responses included in Figure 11.

Unexpectedly an DEmax > 0 is obtained for the three concentrations of THT and the minor concentration 1.96 Vol.% obtains a DEmax = 8.2%. The result indicates that THT acts as initiator in the polymerization reaction since polymer A does not have the initiator NPG. When the concentration is increased, a decreasing of the DEmax is obtained: DEmax = 2.4% for THT 3.85 Vol.% and for 5.7 Vol.%.

It is known that THT can be easily oxidized, including photochemically [32,33]. In the oxidation reaction, the generation of a alpha-carbon thioether radical is favoured [34]. Taking into account that EE is photoreduced during the holographic recording it is possible that THT oxidizes forming a free radical, in a similar way as happens with the initiator NPG for polymer B (Table 1). Thus, THT reacts with excited EE generating THT· radicals according to the overall reaction:(6)$$\mathrm{EE}+\mathrm{THT}\overset{h\nu_{532}}{\to }{\mathrm{EE}}_{reduced}+\mathrm{THT}\cdot$$

The radical THT· obtained reacts with the monomer M starting the polymerization.

An excess of THT can favor the reaction between two THT· radicals, producing a dimeric coupling product that is the main product obtained in other photochemical oxidations that have been described [34]. This explains why the concentrations of THT 3.85 and 5.66 Vol.%. obtain a DEmax lower than the concentration 1.96 Vol.% Therefore for low concentrations of THT the probability of finding two THT· radicals is very small and the initiation is not affected for the coupling reaction.

To evaluate this aspect, a new experiment decreasing the concentrations of THT is made. In addition, only the diffracted beam is measured and the other laser beams are blocked due to the fact that the diffracted beam has a very low light intensity and it is not possible to measure if there is diffused light from transmitted or reflected beams. 

Figure 12 includes the angular scan obtained after the recording of the holograms in three samples with different concentration of THT: 0.196, 0.385, and 1.961 ppm.

The lowest concentration of THT is close to the minimum value that can be distinguished from the noise with the radiometer used.

As expected, with a low interval of concentrations the value of DEmax increases with the concentration of THT contrary to what happened in the previous experiment with high concentrations of THT.

Representing the values of DEmax as a function of the concentration of THT, the graph shown in Figure 13 is obtained.

A linear increase of the DEmax with the THT concentration is obtained. This confirms the action of THT as initiator of the polymerization and the low influence of the coupling reaction for this interval of THT concentrations. However, the proposed mechanism is only a hypothesis that must be corroborated with specific instrumental techniques.

## 5. Conclusions

We have demonstrated that TBT and THT are compatible with the composition of a holographic material type HPDLC and they interfere in the photopolymerization reactions during the holographic recording. TBT interferes in the initiation stage and acts as a retarder with a catalytic mechanism in which a small proportion of TBT has a great influence on the reduction of diffraction efficiency. THT acts as an initiator in a concentration lower than 2 Vol.%. A relatively high concentration produces a quick decrease in its effectiveness as initiator. In addition, the maximum diffraction efficiency obtained is directly proportional to the concentration of THT in the concentration interval [0.2–2] ppm. This result is interesting in the determination of low concentrations of THT. 

The actions that produce both substances are opposite. THT favors the polymerization in the absence of NPG and TBT hinders the polymerization in the presence of NPG. Therefore, it is possible to obtain specificity using the same chemistry. Moreover these actions occur for low concentrations, which could be suitable for sensors. 

The concentration of odorants in natural gas is typically 10 ppm or less [35]. This value varies depending on the mixture of substances used and the legislation of each country. In the United States, it is usually an injection rate equivalent to 4 ppm of odorant in the gas stream, although the minimum concentration that could be used is lower [36]. Minimal odorant concentration represents the odorant content in natural gas which fulfills the requirement for creating warning odor level. This concentration is about 1 ppm for THT and 0.3 ppm for TBT [37]. 

Commercial sensors for natural gas are based on the catalytic oxidation of methane. Sensors that specifically detect TBT or THT are not used. In the bibliography there are few works related to specific sensors for these substances. For TBT, there is a work that develops a cataluminiscent sensor based on V_2_O_5_ that has a linear response in a range of 5.6–196 μg/mL with a limit of detection of 0.6 μg/mL [38]. For THT, there is described a chromatographic method, not a sensor, that uses preconcentration techniques having a limit of detection of 2.0 μg/m^3^ [39]. The use of preconcentration techniques is a strategy that could be used in our holographic method and could allow us to increase the sensitivity decreasing the minimum detectable concentration.

Another aspect to be highlighted is that in the range of concentration studied, the variation of DEmax as a function of the concentration shows an exponential tendency for TBT and linear for THT although this must be confirmed with additional experiments. This result adds interest for its application in a technique of detection through the development of a sensor that can measure both substances.

Finally, a possible mechanism has been suggested regarding the influence of TBT and THT in the chemical reaction scheme that occurs during photopolymerization. This mechanism is only a hypothesis that must be corroborated with instrumental techniques.

The next step would be to work with these substances in gas phase. For this, it is necessary to work on the photopolymer and the preparation of the sample with the objective that the material can absorb TBT and THT from the atmosphere.

## Figures and Tables

**Figure 1 polymers-11-00254-f001:**
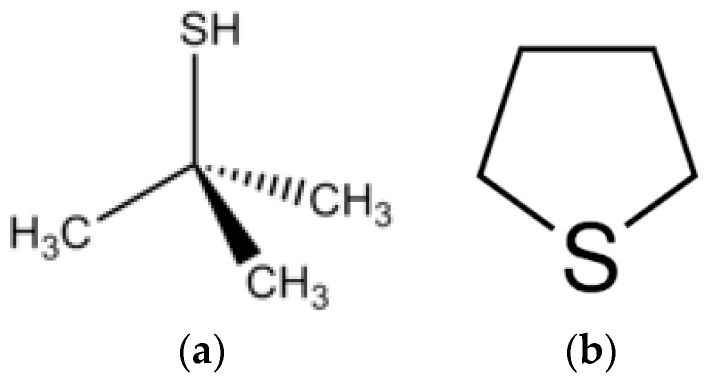
Molecular structure of (**a**) tert-Butylthiol (TBT) and (**b**) tetrahydrothiophene (THT).

**Figure 2 polymers-11-00254-f002:**
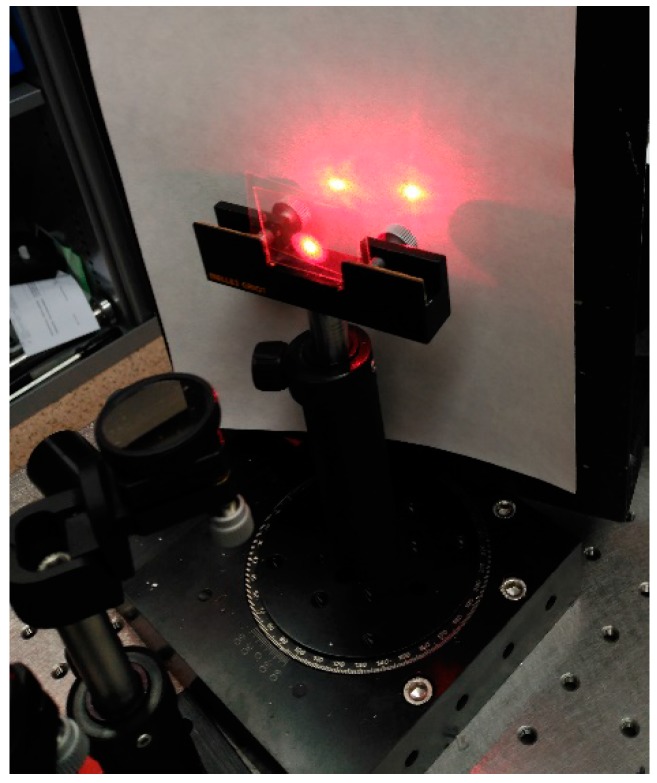
Beams of light diffracted and transmitted by the hologram and diffused light mainly due to the polyester layers.

**Figure 3 polymers-11-00254-f003:**
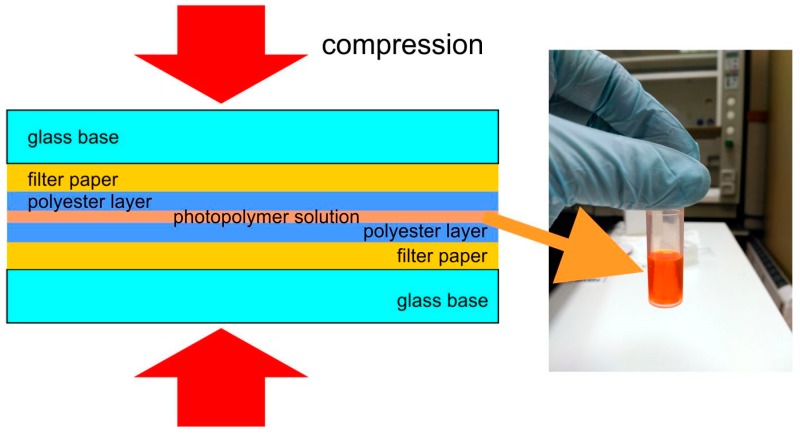
Preparation of the sample by compression of the photopolymer solution inside the layer sandwich.

**Figure 4 polymers-11-00254-f004:**
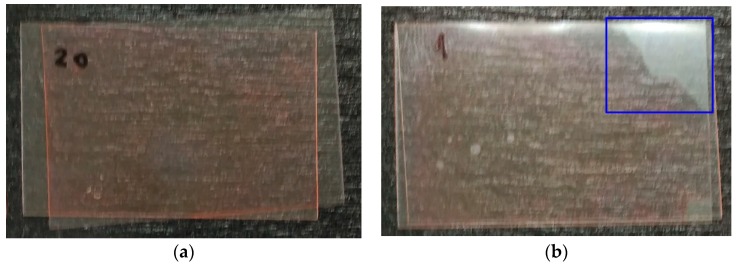
(**a**) Example of homogeneous photopolymer layer on the polyester surface. (**b**) Example of non-homogeneous photopolymer layer with an uncovered area inside the blue rectangle on the polyester surface.

**Figure 5 polymers-11-00254-f005:**
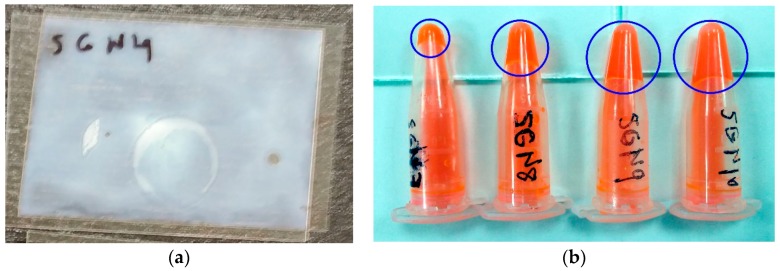
(**a**) Opaque sample by arrangement of LC molecules due to low ambient temperature. (**b**) Polymerized solution inside the blue circle due to high temperature during the preparation of the solutions.

**Figure 6 polymers-11-00254-f006:**
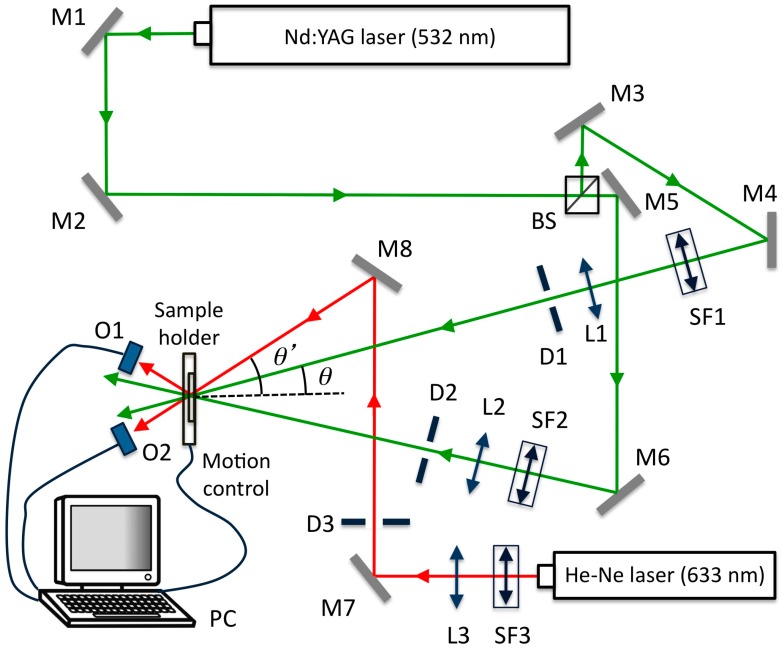
Optical set-up. BS: Beamsplitter, Mi: mirror, SFi: spatial filter, Li: lens, Di: diaphragm, Oi: optical power meter, PC: data recorder.

**Figure 7 polymers-11-00254-f007:**
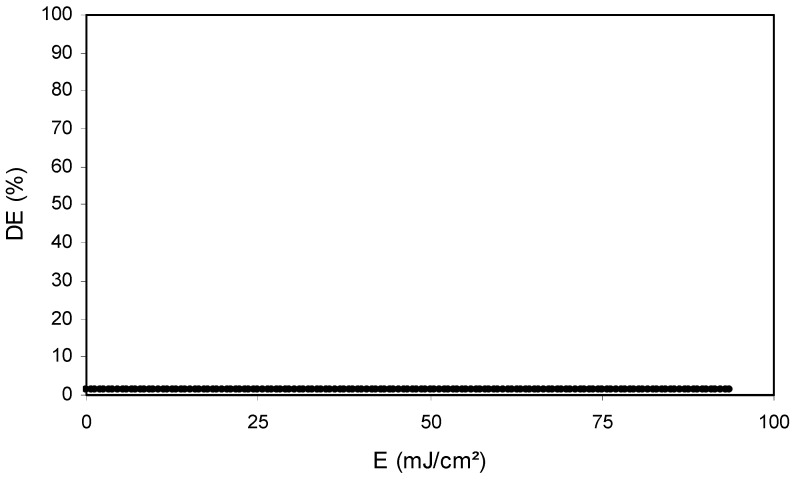
Diffraction efficiency versus energetic exposure for a sample with polymer A and TBT.

**Figure 8 polymers-11-00254-f008:**
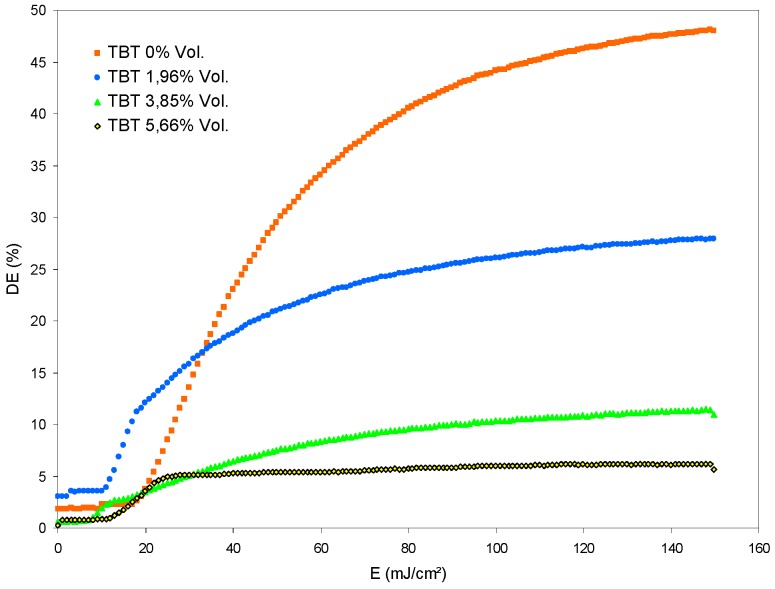
Diffraction efficiency versus energetic exposure for samples with polymer B and different concentrations of TBT.

**Figure 9 polymers-11-00254-f009:**
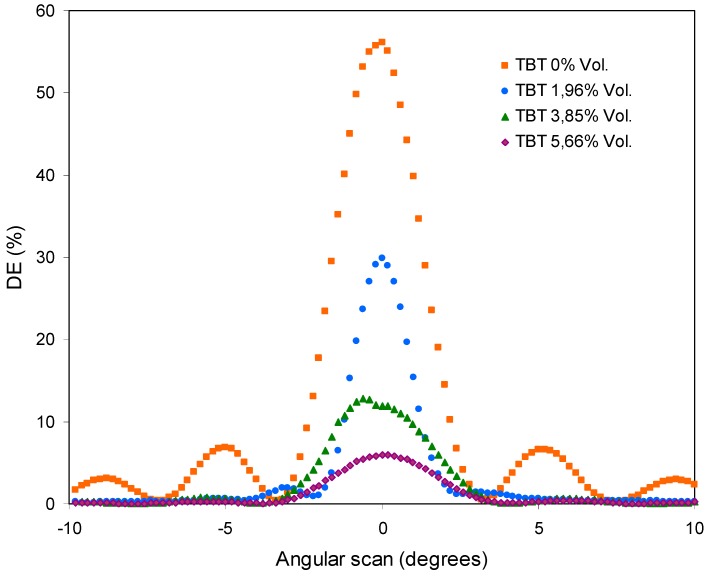
Diffraction efficiency versus angular scan for samples with polymer B and different concentrations of TBT.

**Figure 10 polymers-11-00254-f010:**
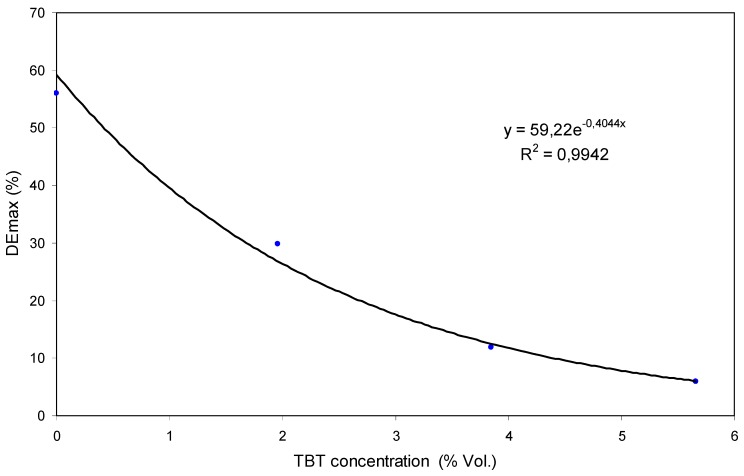
Maximum diffracted efficiency (DEmax) values from Figure 9 versus TBT concentration.

**Figure 11 polymers-11-00254-f011:**
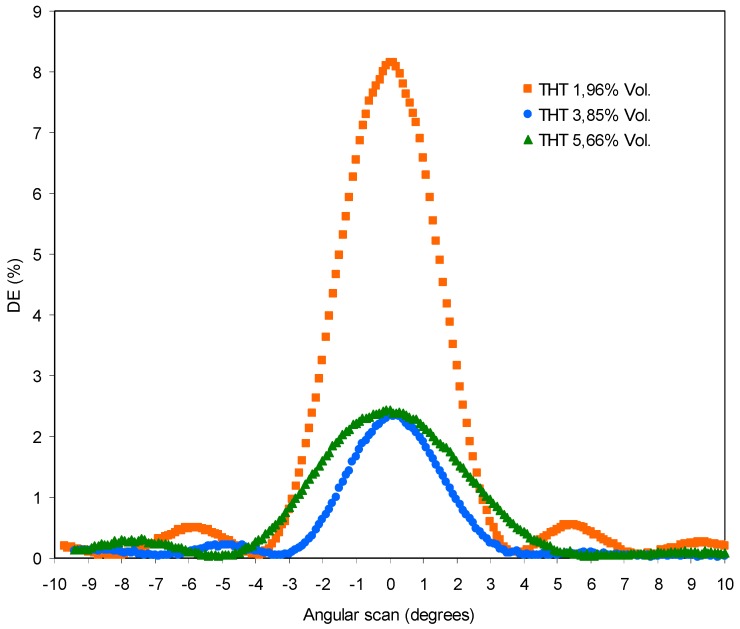
Diffraction efficiency versus angular scan for samples with polymer A and different concentrations of THT.

**Figure 12 polymers-11-00254-f012:**
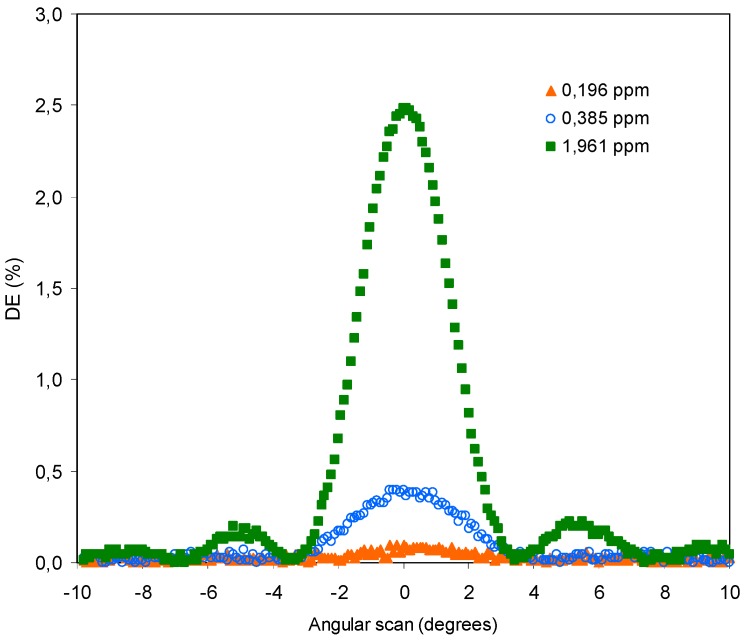
Diffraction efficiency versus angular scan for different concentrations of THT in the photopolymer in parts per million.

**Figure 13 polymers-11-00254-f013:**
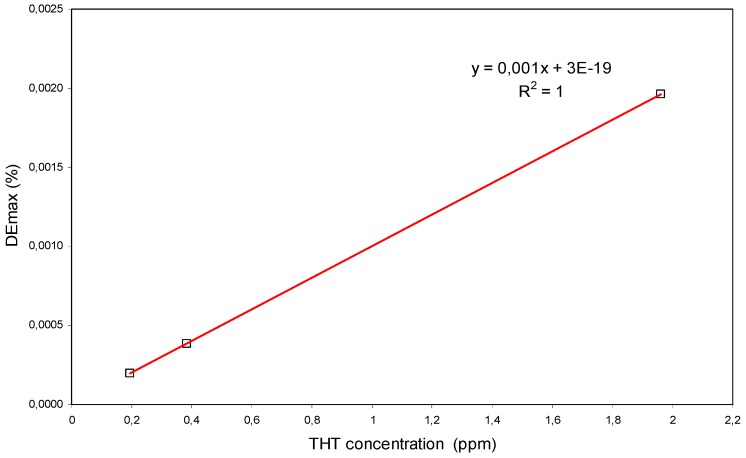
DEmax versus concentration of THT.

**Table 1 polymers-11-00254-t001:** Composition for liquid crystal-photopolymer composites.

Component	Polymer A Concentration (wt %)	Polymer B Concentration (wt %)
DPHPA	44.27	44.10
QYPDLC-036	30.43	30.31
EE	0.04	0.04
NPG	0.00	0.39
OA	8.79	8.76
NMP	16.47	16.40

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
