# Peer review of "Influence of Tert-Butylthiol and Tetrahydrofuran on the Holographic Characteristics of a Polymer Dispersed Liquid Crystal: A Research Line Toward a Specific Sensor for Natural Gas and Liquefied Petroleum Gas"

_polymers, 2019, doi:10.3390/polym11020254_

Round 1

Reviewer 1 Report

This paper presents a new sensor for tert-butylthiol (TBT) and tetrahydrothiophene (THT) incorporeted into a holographic polymer-dispersed liquid crystal (PDLC) system. The main and original part of the work is related to the preparation of the samples and the optical measurements, with the aim to investigate the diffraction efficiency and diffraction properties of the new sensor in presence of different concentrations of TBT and THT.

The results reported in the paper, however, are not very satisfying, in particular, the authors have reported only three values of concentrations for THT and four values of concentrations for TBT. Few more points would be important to better understand the trend of the diffraction efficiency versus the concentration and to discriminate between a linear and a non-linear trend.

Moreover, a comparison between these results and other analytical methods used to detect TBT and THT molecules (including references from the literature) would be important to understand the significance and applicability of the new sensor.

Author Response

comments to reviewer 1:

We appreciate the comments of the reviewer to improve the manuscript.

"...The main and original part of the work is related to the preparation of the samples and the optical measurements..."

The preparation of the samples and the optical measurements are mandatory requirements to ensure the correct interpretation of the results, but they are not the main and original part of this work. In this work, we are used for first time TBT and THT as components of a HPDLC material. We have not found in the bibliography until 2018 any previous work that uses TBT and THT as components of a HPDLC photopolymer.

The original part of this work is that TBT and THT interfere the photopolymerization reactions that take places in the material used as holographic recording media. Moreover they act in a completely differently way. This original result is compatible with the objective of achieving a holographic sensor or at least a specific detection holographic method for TBT and THT.

This is what must be valued in this work regardless of whether the method of preparing the samples has been described in detail.

"...The results reported in the paper, however, are not very satisfying, in particular, the authors have reported only three values of concentrations for THT and four values of concentrations for TBT. Few more points would be important to better understand the trend of the diffraction efficiency versus the concentration and to discriminate between a linear and a non-linear trend."

This work involves the opening of a new line of research and its objective is not a precise mathematical determination. The main result of this research is that TBT decreases the polymerization in the HPDLC composite, even acting as a retarder and THT acts as initiator of the polymerization.

We agree with the reviewer that more points will be needed to better determine the correct mathematical relationship between the variables involved and we will undoubtedly do so in future work when the experimental procedure allows us to prepare the samples faster. Currently for us three values of concentrations for THT and four values of concentrations for TBT have meant much more experimental work than what represent 3 or 4 points in the graphs. The number of experimental points is small because in the planning of the experiments we did not expect the good results obtained.

We can not do more experiments and add more points to the graphs that we already have due to several factors:

The preparation of the samples by hand requires a slow process that discards many samples and this implies many time elapsed.

The samples are very sensitive to room temperature. Small changes in room temperature affect the results.

Samples prepared on different days are not suitable for previous experiments.

The photopolymer solution ages with the time elapsed since its preparation and therefore for each new series of experiments, a new photopolymer solution must be prepared

But the main problem is the next:

Complexity of working with TBT and THT and necessary authorizations, since they generate social alarm. Small amounts of TBT and THT (nanograms) are diffused through the ventilation and air conditioning systems and generate an odour that reminds to explosive gas throughout the building, even outside the building.

However, the reviewer will agree with us that although the number of points used is small, the correlation coefficients obtained R2> 0.99 for TBT and THT (Figures 10 and 13) are sufficient to suppose that there is a relationship between the implied parameters that show an exponential behaviour for TBT and linear for THT. Nevertheless the precise mathematical relationship is a secondary aspect for this work.

Moreover, a comparison between these results and other analytical methods used to detect TBT and THT molecules (including references from the literature) would be important to understand the significance and applicability of the new sensor.

We have added this comparison in the text. We have added new references.

Reviewer 2 Report

The Authors present an interesting application of polymeric dispersed LC as holographic sensors for thiols, therefore for natural gas. The presence of thiols interfere with the polimerization reaction, thus affecting the holographic outcome. The results are very interesting and the work appears solid and well conducted. Therefore I believe the paper can be published, however the Authors should address few minor issues in their revised version.

First, the Authors should mention how the sensitivity of their method compares with the concentrations of thiols usually found in natural gas.

Then, the mechanism related with the THT detection is not fully convincing. While for TBT is quite reasonable to assume that a tBU-S radical is formed by hydrogen abstraction, the formation of the THT radical is not so obvious. The Authors cite a paper by Chambers and Hill (Inorg. Chem. 1991), however in that case the THT radical was observed by using a polyoxometallated catalysts and in anaerobic conditions leading to a recombination to form te dimer. It seems that the situation is quite different here so the proposed model is not fully satisfactory.

The Authors should elaborate more on the proposed mechanism.

Author Response

comments to reviewer 2:

We appreciate the comments of the reviewer to improve the manuscript.

"First, the Authors should mention how the sensitivity of their method compares with the concentrations of thiols usually found in natural gas."

We have added information about this aspect in the text

"Then, the mechanism related with the THT detection is not fully convincing. While for TBT is quite reasonable to assume that a tBU-S radical is formed by hydrogen abstraction, the formation of the THT radical is not so obvious."

We use THT with polymer A as described in the text. Polymer A has not initiator and therefore it can not generate free radicals during the laser exposure. In other words, polymer A is not a photopolymer with the laser intensity and exposure time used here.

Photopolymer A with THT obtains a DE>0 in Figures 11 and 12 and it is exclusively due to the presence of THT in the formulation. Therefore THT is the initiator and must initiate the polymerization reaction by means of a radical intermediary. If THT did not produce a free radical we would not obtain DE>0 in Figures 11 and 12. As EE is photoreduced under laser exposure this implies that THT must be oxidized.

We have improved the explanation in the text.

"...The Authors cite a paper by Chambers and Hill (Inorg. Chem. 1991), however in that case the THT radical was observed by using a polyoxometallated catalysts and in anaerobic conditions leading to a recombination to form te dimer..."

We have cited this work because it explains that in an oxidation THT tends to generate a radical in the alpha carbon to the S atom. The oxidation conditions are not the same but the radical derived from THT may be if we take account that the radical in alpha carbon is an intermediary favoured for this molecule.

"...It seems that the situation is quite different here so the proposed model is not fully satisfactory. The Authors should elaborate more on the proposed mechanism."

The proposed model takes into account the photoreduction of EE and the consequent oxidation of THT forming a free radical that initiates the polymerization reaction. It is the most logical alternative that explains the experimental facts: photoreduction of EE forming a colorless derivative and polymerization of polymer A that does not contain initiator. In any case it is a tentative mechanism as indicated in the text.

We have added a new reference related to this:

E. Campaigne "Comprehensive Heterocyclic Chemistry" Volume 4, 1984, Pages 863-934. Cap. 3.15 - Thiophenes and their Benzo Derivatives: (iii) Synthesis and Applications

Here is cited that tetrahydrothiophenes are easily oxidized to thiophenes. In THT this oxidation reaction can occur through the formation of a radical intermediate in the alpha carbon.

The obtaining of a dimeric coupling product formed by two THT radicals when there is an excess of THT radicals is a very plausible assumption independently that Chambers obtained those free radicals in another way. Result from experiment in Figure 12 is according to this hypothesis.

To propose a more elaborate reaction mechanism or corroborate the proposed it would be necessary to use analytical techniques, capture radicals, etc. But that is not the goal of this work and the proposed mechanism is only a hypothesis according to the obtained results and the chemical characteristics of the substances involved. This has been underlined in the text.

Round 2

Reviewer 1 Report

The authors have addressed most of the points. Despite of the main experimental critisisms have not been solved, due to technical problems, several issues have been commented and the paper is more acceptable for publication.